# Application of Cytokinin under Modified Atmosphere (MA) Delays Yellowing and Prolongs the Vase Life of *Davallia solida* (G. Forst.) Sw. Leaves

Pattaraporn Ngamkham [1], Varit Srilaong [1,2], Chalermchai Wongs-Aree [1,2] and Mantana Buanong [1,2,*]

1 Division of Postharvest Technology, School of Bioresources and Technology, King Mongkut's University of Technology Thonburi, Bangkok 10140, Thailand
2 Postharvest Technology Innovation Center, Ministry of Higher Education, Science, Research and Innovation, Bangkok 10400, Thailand
* Correspondence: mantana.bua@kmutt.ac.th; Fax: +662-470-3752

**Abstract:** Cut leaves of *Davallia solida* are widely used in bouquet greenery. However, the leaves wilt and yellow after seven days. Postharvest applications of exogenous cytokinins (CKs), a plant growth regulator (PGR), preventing senescence in many green leafy plants, were studied by pulsing cut *D. solida* leaves with 6-benzylamiopurine (BA) at a concentration of 100 ppm, or thidiazuron (TDZ) at a concentration of 10 μM for 24 h, compared with distilled water as a control, and then placing the ferns in a controlled room (21 + 2 °C, 70–80% relative humidity (RH), under cool-white fluorescent lights for 12 h/d). Pulsing with BA and TDZ delayed leaf yellowing while preserving chlorophyll (Chl) content. This was due to reduced Chl-degrading enzyme activities on day 8 of the vase life of the leaves, resulting in longer display life of 11.1 and 11.5 days, respectively. TDZ delayed Chl breakdown on day 8 of the vase life of leaves more than was the case for BA. Subsequently, leaves were pulsed in 10 μM TDZ, or distilled water for 24 h, and then 10 leaves from each treatment were packaged in a 25 × 30 cm, 25-μm thickness BOPP bag. All the packages were stored at 10 °C in 10 h/d light for one, two and three weeks, then unpacked and placed in distilled water. Pulsing with TDZ before one-week storage delayed leaf yellowing, maintained Chl content and extended their vase life to 9.6 days compared with one-, two-, and three-week control leaves, which had a vase life of 6.2, 5.8, and 3.6 days, respectively. After one week, the relative fresh weight of the leaves and water uptake did not alter between the control and TDZ groups. The activities of Chl degradation enzymes in the leaves of *D. solida* pulsed with TDZ prior to storage were significantly suppressed, related to higher Chl content and a lower percentage of leaf yellowing than the control, resulting in a longer vase life of 9.0 days, while the control lasted 7.0 days. A 10 μM TDZ pulsing treatment significantly prevented the yellowing of *D. solida* fronds during the vase period or storage period, and one-week storage under MA with light conditions is recommended for retaining postharvest quality after storage.

**Keywords:** rabbit's foot fern; TDZ; chlorophyll; postharvest longevity; storage

## 1. Introduction

Florists' greens are widely used as fillers in ornamental bouquets in Thailand to increase the value of flower products to consumers. Although the vase life of florists' greens is usually long enough to satisfy customers' needs, the storage period needs to be extended for as long as possible to cover periods of product shortage [1]. Ferns are used as florists' greens and are grown worldwide, including in Thailand, because of their ready availability, high consumer acceptance, low cost and good longevity [2,3]. *D. solida* is one of the most commercial ferns sold in flower markets, due to the beautiful, delicate and complex symmetry of its glossy green foliage, but its vase life is rapidly short within seven days resulting from leaf yellowing, and/or desiccation (wilting) [4,5]. Kowwilaisaeng and Teerarak [6] reported that the senescence of *D. solida* ferns was related

to the increase in malondialdehyde (MDA) and $H_2O_2$ content, and the decrease of fresh weight, Chl and carotenoid contents. The loss of ornamental value of leafy plants such as yellowing, drooping, wilting or withering greatly varies between species and cultivars, which affects postharvest longevity, resistance to transport conditions and storability [7]. Leaf senescence is involved in the degradations of proteins, Chl, nucleic acids, membranes and the subsequent transport of some degrading products to other plant parts [8,9]. The yellowing of leaves resulting from Chl degradation is the most obvious visible appearance. During the process of leaf senescence, Chl degradation in plants has been elucidated as the pheophorbide *a* oxygenase (PaO) pathway [10,11]. In many plants, the removal of the phytol moiety from the Chl molecule by chlorophyllase (Chlase) is the first step to forming the first green breakdown product, chlorophyllide (Chlide) *a*. Then, the central $Mg^{2+}$ from chlide is removed by Mg-dechelatase (MD) and the green color of Chl catabolites is completely lost when the porphyrin ring of pheophobide (Pheide) *a* is cleaved by pheophorbide *a* oxygenase (PAO) as a result of oxidized red Chl catabolite (RCC), which is subsequently catalyzed by red Chl catabolite reductase (RCCR) to generate primary fluorescent Chl catabolite (pFCC) [12–16]. The final step is the modification of pFCC, which is transported into the vacuole, and isomerized to non-fluorescent products by acidic pH [17–19]. Moreover, Chl-degrading peroxidase (Chl-POX) is thought to be involved in the degradation of Chl *a* in the first step to forming $13^2$-hydorxy chlorophyll (C$13^2$-OHChl) [20,21]. Additionally, pheophytinase (pheophytin pheophorbide hydrolase, PPH) is a new Chl-degrading enzyme which dephytylates the Mg-free Chl and pheophytin (Phein) *a* to Pheid [22,23].

Some postharvest methods applied as preservative solution are effective for retaining freshness and retarding the yellowing of cut leaves of *D. solida* [24]. Cytokinins (CKs) markedly delay or reverse leaf yellowing and senescence in various species, as well as promoting growth processes in plants [25–27]. Asami and Nakagawa [28] reported that during the progressive ageing process in plant tissues, the levels of CKs and gibbellerins (GAs), an inhibitor of the ageing process, decreased, while the levels of accelerating ageing regulators such as ethylene, salicylic acid (SA), brassinosteroids (BR), abscisic acid (ABA) and jasmonic acid (JA) increased. CK signal transduction is perceived by the receptors histidine kinases (AHK2, AHK3) and transduced to the B-type transcription factors Arabidopsis response regulators (ARR2) and cytokinin response factors (CRFs) [29]. ARRs directly regulate the expression of genes in Chl biosynthesis and the light harvesting complex, such as HEMA1 and LHCB6 [30,31], suggesting that CKs can increase Chl content and delay leaf senescence. Therefore, the external application of CKs significantly retarded the senescence process and extended the postharvest longevity of many species of florists' greens such as *Alchemilla mollis* [32], *Limonium latifolium* [33], *Arum italicum* [34]. Benzyladenine (BA), one of the synthetic CKs, is the most commonly used in floriculture production and effectively prolongs the postharvest life of many species cultivated for florists' greens [7]. However, the efficiency of these compounds depends on the mode of application, as well as on the type and concentration of CK used [35]. Moreover, thidiazuron (TDZ; *N*-phenyl-*N*′-1,2,3-thiadiazol-5-ylurea) is a non-metabolizable phenylurea compound with strong cytokinin-like activity [30] and has been commonly used at either high (e.g., 100 µM) concentrations as a defoliant or low (e.g., 1 µM) concentrations for regeneration in plant tissue culture [36,37]. A previous study showed that cytokinin-active phenylureas, such as TDZ and 4-PU-30 (N-phenyl-N′-(2-chloro-4-pyridyl)urea), can elicit many biological properties qualitatively similar to those of adenine-type CKs, but also some quite different properties [38]. Treatment with TDZ has also been reported to retard leaf senescence in cut flowers of *Chrysanthemum*, *Alstroemeria* and *Tulipa* [39–41], to reduce flower abscission and the senescence in cut inflorescences of *Phlox paniculata* and *Lupinus densiflorus* [42,43], and to delay leaf yellowing in pot plants of *Pelargonium hortorum* 'Tango', *Freesia*, *Ornithogalum* and *Euphorbia fulgens* [44]. Tamala et al. [5] reported that holding treatment with TDZ at a concentration of 10 µM significantly prolonged the vase life of cut leaves of *D. solida* by 1.5 days longer than the control. However, pulsing treatment with TDZ has not been tested on the leaves of *D. solida*. If effective, the treatment could

be applied by the producers of the ferns for improving the quality and prolonging the postharvest life of *D. solida* leaves. Therefore, the objective of this study was to investigate whether or not the pulsing treatment with CKs could delay leaf yellowing of *D. solida* after harvest, and whether the application of CKs under MA could extend the storage life and vase life of leaves of this species.

## 2. Materials and Methods

### 2.1. Plant Materials and Treatments

Leaves of *Davallia solida* (G. Forst.) Sw. were harvested in the morning, washed and delivered to a flower market. Leaves at the commercial stage with no spores (25 cm × 40 cm) as shown in Figure 1 were obtained from Pak Klong Talad (flower market), Bangkok, and transported to the laboratory at King Mongkut's University of Technology Thonburi (KMUT) within 2 h. After arrival at the laboratory, those showing fresh greenish leaves, of the same size and without any defect were selected, then re-cut under water to 12 cm leaf petiole length and placed in buckets with distilled water.

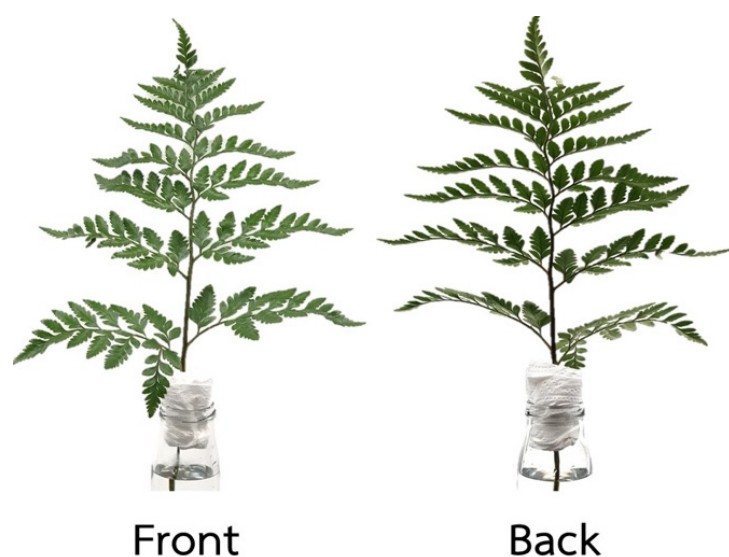

**Figure 1.** Cut leaves of *D. solida* used in the experiment.

### 2.2. Cytokinin Pulsing Treatments

2.2.1. Effect of CKs Pulsing on Delaying Leaf Yellowing and Extending the Vase Life of *D. Solida* Leaves

The leaves were then divided into three groups for pulsing with 250 mL TDZ (10 μM), BA (100 ppm = 444 μM) (Sigma-Aldrich GmBH, Germany) [45] and distilled water as the control for 24 h, under a room temperature of 21 ± 2 °C and 70–80% RH. After pulsing, they were transferred to 250 mL distilled water and kept in a controlled environment room: 21 + 2 °C, cool-white fluorescent lights for 12 h/d, 60–70% RH.

2.2.2. Simulations of the *D. solida* Leaves Vase Life after CKs Pulsing Treatment and Storage in a Modified Atmosphere Packaging (MAP)

The leaves were pulsed with 250 mL TDZ (10 μM) and distilled water as the control for 24 h, 21 ± 2 °C. After pretreatments, MAP experiments were conducted by placing samples of 10 leaves in a 30 cm × 45 cm biaxially oriented polypropylene (BOPP) bag with 25-μm thickness, which was tightly sealed by a tabletop vacuum sealer (Fuji Impulse Co., Ltd., Osaka, Japan). All the leaves were stored at 10 °C for 1, 2 and 3 weeks in light conditions of 12 h/day, as used for tropical green foliage storage, then subsequently unpacked and transferred to distilled water for the duration of the experimental period in a controlled environment room: 21 + 2 °C, 70–80% RH under cool-white fluorescent light for a 12-h photoperiod.

### 2.2.3. Optimal Pre-storage Conditions of *D. solida* on Activities of Chlorophyll-Degrading Enzymes

*D. solida* leaves were pretreated in the same way as mentioned above, but were stored at 10 °C for 1 week in light conditions of 12 h/day, as used for tropical green foliage storage, then subsequently unpacked and transferred to distilled water for the experimental period in a controlled environment room: 21 + 2 °C, 70–80% RH under cool-white fluorescent light for a 12-h photoperiod.

### 2.3. Measurement of Fresh Weight and Water Uptake

The relative fresh weight of the leaves was taken individually every 2 days throughout the experimental period. The changes in fresh weight were presented as a percentage of the initial weight of leaves. Water uptake was measured from the volume losses of the vase solutions from 10 mL of solution in test tubes every 2 days throughout the experimental period.

### 2.4. Measurement of Yellowing and Vase Life of Leaves

Leaf yellowing was observed by calculating the number of yellowing leaves on the stem as the percentage of leaf yellowing. The vase life of the leaves was determined as the period from the start of experiment to the time when chlorosis (yellowing) and/or desiccation (wilting) leaves were more than 30%.

### 2.5. Measurement of Chlorophyll Content

The Chl content in leaves was measured by exaction with *N,N*-dimethylformamide in darkness at 4 °C for 24 h. The absorbance reading was performed at 647 and 664 nm and then calculated as described by Moran [46].

$$\text{Total Chl (Chl}_{a+b}) (\mu L \cdot mg \text{ of plant extract}) = (20.2A_{645} + 8.02A_{663}) \times wl \tag{1}$$

$C_a$ = Chl *a*
$C_b$ = Chl *b*
wl = weight loss (%)

### 2.6. Measurement of Activities of Chlorophyll-Degrading Enzymes

*Preparation of acetone powder [47]*

Pigments were eliminated by grinding 5 g of tissue samples in 50 mL of 80% cold acetone with a homogenizer, then filtered with Whatman no. 1. The residue (acetone powder) was rewashed with cold acetone until it became colorless. The combined supernatant was brought to 50 mL diethyl ether and dried in a desiccator for 3 h. The acetone powder was kept at −20 °C for enzyme activity analysis.

*Preparation of chlorophyll a*

*Chl a* substrate was prepared from 15 g fresh leaves according to Costa et al. [48]. The leaves were homogenized in 60 mL cold acetone (−20 °C) for 3 min then filtrated through two layers of Miracloth (Calbiochem®, Darmstadt, Germany). The extract was then purified by dioxane precipitation. The ratio of dioxane: acetone was 1:6 (*v/v*). Then, 15 mL of distilled water was added to the extract and kept in darkness at 4 °C for 1 h. Next, the filtrates were centrifuged at 10,000× *g* for 15 min at 4 °C. The precipitate was resuspended with 40 mL of 80% acetone, 6 mL dioxane and 14 mL distilled water and centrifuged at 10,000× *g* for 15 min at 4 °C, and subsequently dissolved in petroleum ether. The soluble Chl in petroleum ether was stored at −20 °C until the pigments were individually separated using sugar powder column chromatography [49]. Finally, 500 μg·mL$^{-1}$ of Chl *a* was prepared in acetone.

*Preparation of chlorophyllin a*

Chlorophyllin (Chlin) *a* was slightly modified according to Suzuki and Shioi [50]. The Chl *a* acetone solution (3 mL) was partitioned into petroleum ether. The petroleum ether phase was washed three times with 20 mL of distilled water, after which 2 mL of 30% (*w/v*) KOH in methanol was mixed in the solution. The Chlin *a* was allowed to precipitate and then centrifuged at $16,000 \times g$ at 4 °C for 10 min. The precipitate was dissolved in distilled water, brought to pH 9 with 2 M tricine and stored in the dark at −20 °C. Chlin was used as the substrate for the Mg-dechelatase assay.

*Preparation of pheophytin a*

Pheophytin *a* was slightly modified using the method of Schelbert et al. [22]. A 3 mL Chl *a* acetone solution was added using a few drops of 0.1 N HCl and 0.1 N NaOH.

*Analysis of chlorophyll degrading enzyme activities*

An acetone powder (0.5 g) of tissue samples was suspended in 15 mL 10 mM phosphate buffer (pH 7.0) containing 0.6% CHAPS for chlorophyllase (Chlase), in 15 mL 50 mM phosphate buffer (pH 7.0) containing 50 mM KCl and 0.24% Triton-X 100 for Mg-dechelatase (MG), in 15 10 mM phosphate buffer (pH 7.0) for chlorophyll-degrading peroxidase (Chl-POX), or in 15 mL 50 mM Tris-HCl (pH 8.0) for Pheophytinase (PPH). The crude enzyme was stirred for 1 h at 0 °C and the mixture was filtered with two layers of Miracloth. The filtrate was then centrifuged at $16,000 \times g$ at 4 °C for 15 min. The supernatant was used as the crude enzyme extract. The enzyme protein contents were determined using Bradford's method [51].

*Chlorophyllase activity (Chlase)*

Chlase activity was modified using the method described by Harpaz-Saad et al. [52]. The reaction mixture contained 0.5 mL of enzyme extract, 0.5 mL 0.1 M phosphate buffer (pH 7.5) containing 1.44% Triton X-100, and 0.2 mL Chl acetone solution. The reaction mixture was incubated at 25 °C for 40 min, and terminated by the addition of 4 mL acetone. Chlide *a* was separated from Chl *a* by adding 4 mL of hexane at which the remaining Chl *a* was the upper phase and Chlide *a* was the lower aqueous layer. Chlase activity was spectrophotometrically assayed by Chlide *a* formation at 667 nm per unit per mg protein. The amounts of Chlide generated were determined using extinction coefficients of 76.79 $mM^{-1} \cdot cm^{-1}$ [39] and the enzyme activity was represented as unit·$mg^{-1}$ protein.

*Chlorophyll-degrading peroxidaseactivity (Chl-POX)*

Chl-POX was determined as previously described [53]. The reaction mixture contained 0.5 mL of enzyme solution, 0.1 mL 1.44% Triton-X 100, 0.1 mL 5 mM p-coumaric acid, 0.2 mL 500 $\mu g \cdot ml^{-1}$ Chl *a* acetone solution, 1.5 mL 0.1 mM phosphate buffer (pH 5.5) and 0.1 mL 0.3% hydrogen peroxide. The activity was monitored spectrophotometrically by the measurement of the decrease of Chl *a* at 668 nm per unit per mg protein at 25 °C.

*Mg-dechelatase activity (MD)*

Mg-dechelatase activity using chlin *a* was modified by the method of Suzuki and Shioi [50]. The activity was spectrophotometrically determined by monitoring the absorbance of pheophorbin *a* formation at 686 nm [54]. The assay mixture contained 0.75 mL of 50 mM Tris-Tricine (pH 8.0), 0.3 mL of Chlin *a* and 0.5 mL of enzyme solution, and was incubated was incubated at 37 °C for 3 min.

*Pheophytinese activity (PPH)*

Pheophytinase activity using pheophytin *a* was modified from Schelbert et al. [22]. The reaction mixture contained 0.5 mL enzyme solution, 0.5 mL 15 mM Tris-HCl (pH 8.0) and 0.2 mL pheophytin a, and was incubated at 25 °C for 30 min. The reaction was ended by the addition of 2 mL of acetone. Pheophocide *a* was separated from the pheophytin a by adding 2 mL of hexane at which point the lower aqueous layer was spectrophotometrically assayed at 667 nm per unit per mg protein. The amounts of pheophocide *a* generated were determined using extinction coefficients of 7.08 $mM^{-1} \cdot cm^{-1}$ and the enzyme activity was represented as unit·$mg^{-1}$ protein.

### 2.7. Experimental Design and Statistics Analysis

Experiments 1 and 3 were conducted in a completely randomized design (CRD) using 8–10 individual leaf stems per treatment and experiment 2 was conducted in factorial in completely randomized design using 8–10 individual *D. solida* leaves per treatment (FW, water uptake, percentage of leaf yellowing and vase life), and Chl content and enzymatic parameters (activities of Chlase, MG, Chl-POX and PPH enzymes) were analyzed using three replicate samples by using a whole leaf. The data obtained were subjected to analysis of variance (ANOVA) and differences between the means were compared by using Duncan's multiple range test (DMRT) and a *t*-test in Excel.

## 3. Results

### 3.1. Effects of Cytokinins on Chlorophyll Content and Display Quality of D. solida

The Chl content in *D. solida* ranged from 42.73 to 93.90 mg·100 g$^{-1}$ FW and had a trend towards declining during the vase period (Figure 2A). The level of total Chl in the control leaves increased to the highest content of 93.90 mg·100 g$^{-1}$ FW on day 2, then sharply decreased to 45.09 mg·100 g$^{-1}$ FW on day 12 of the vase life, while pulsing treatments with CKs significantly maintained the Chl content in the *D. solida* leaves during the vase period. The Chl content of the *D. solida* leaves pulsed with TDZ and BA slightly declined during the first 6 days of vase life and increased thereafter to 70.11 and 65.82 mg·100 g$^{-1}$ FW, respectively on day 12. Chl *a* had a similar trend to total Chl content, ranging between 34.91 and 63.24 mg·100 g$^{-1}$ FW. The content of Chl *a* in the control leaves increased to the highest level of 63.24 mg·100 g$^{-1}$ FW on day 2, then decreased rapidly to 36.61 mg·100 g$^{-1}$ FW on day 12 of the vase period, while the Chl *a* content of the *D. solida* leaves pulsed with TDZ and BA was 44.49 mg·100 g$^{-1}$ FW on the initial day and gradually reduced to 36.87 and 35.73 mg·100 g$^{-1}$ FW, respectively, then rose again to 47.50 and 44.56 mg·100 g$^{-1}$ FW, respectively on day 12 (Figure 2B). However, Chl *b* of cut leaves of *D. solida* had minor changes between 8.48 and 14.09 mg·100 g$^{-1}$ FW during the vase period (Figure 1C). The treatment with CKs showed a higher content of Chl *b* of 14.41 mg·100 g$^{-1}$ FW in TDZ treatment and of 13.14 mg·100 g$^{-1}$ FW in BA treatment, while the control leaves had 11.16 mg·100 g$^{-1}$ FW on day 12.

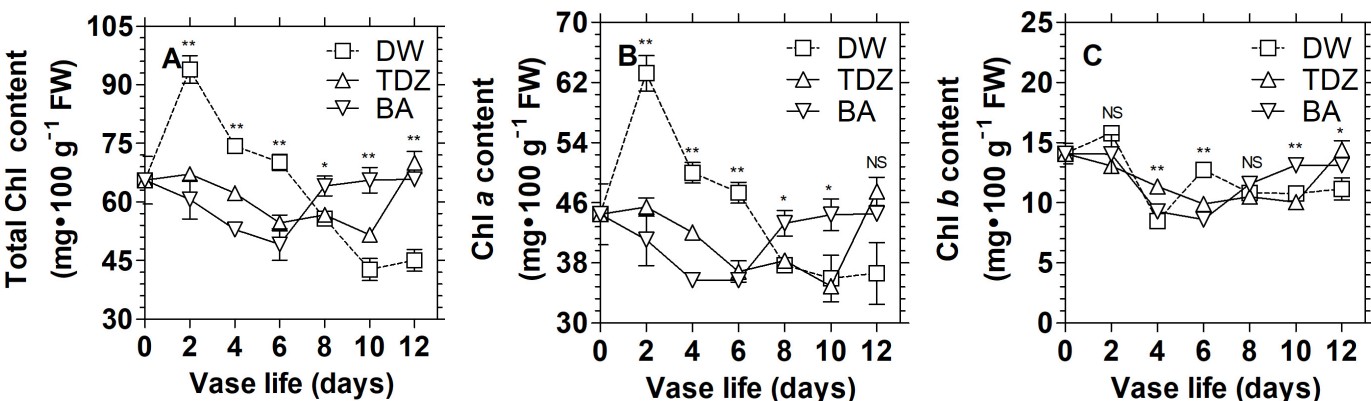

**Figure 2.** Effect of CKs on the contents of total Chl (**A**), Chl *a* (**B**) and Chl *b* (**C**) in *D. solida* leaves. Leaves were pulsed with distilled water (DW; control), 10 μM TDZ, 100 ppm BA for 24 h, then transferred to distilled water in a controlled environment room for the vase period. Asterisks above the curves at each time point represent significant differences between treatments according to Duncan's multiple range test (DMRT); *p* < 0.01 (**) and *p* < 0.05 (*), NS not significant.

The content of Chl was closely related to the percentage of leaf yellowing, ranging between 0 and 41.11% (Figure 3A). The *D. solida* leaves became yellow with time in the vase. The percentage of leaf yellowing of the control continuously rose from day 4 to 41.11% on day 12, which was higher than that of the CK treatments. The pulsing treatments with TDZ and BA delayed the yellowing of leaves during the experimental period. The termination

of the vase life of the leaves was considered to be at 30% of leaf yellowing, as shown in Figure 3B. It was found that the leaves pulsed with CKs had an 11-day vase life, longer than the control leaves, which had a 9-day vase life.

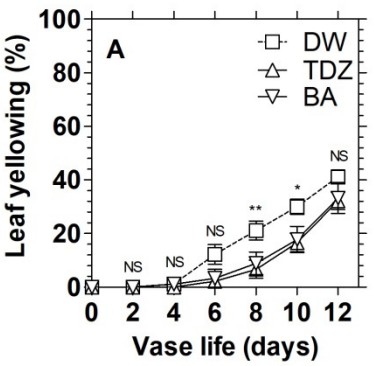 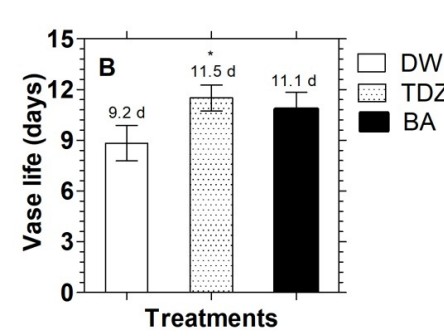

**Figure 3.** Effect of CKs on yellowing (**A**) and vase life (**B**) of *D. solida* leaves. Leaves were pulsed with distilled water (DW; control), 10 μM TDZ, 100 ppm BA for 24 h, then transferred to distilled water in a controlled environment room for the vase period. Asterisks above the curves at each time point represent significant differences between treatments according to Duncan's multiple range test (DMRT); $p < 0.01$ (**) and $p < 0.05$ (*), NS not significant. The numbers above columns represent the values of vase life days.

### 3.2. Effects of Cytokinins on the Activities of Chlorophyll Catabolism Enzymes in D. solida

The activity of Chlase ranged from approximately 247.80–454.93 unit·mg$^{-1}$ protein and tended to increase continuously throughout the vase period (Figure 4A). However, no significant difference was observed in Chlase activity between the treatment types. Chl-POX activity in leaves ranging between 14.54 and 27.26 unit·mg$^{-1}$ protein had a slight change during the vase period, as shown in Figure 4B. The results revealed that the activity of Chl-POX of *D. solida* leaves pulsed with TDZ and BA was almost constant throughout the vase period, while that of the control leaves sharply rose to the highest value of 27.26 unit·mg$^{-1}$ protein on day 8 and decreased thereafter. In addition, the result showed that the activity of Chl-POX in leaves treated with CKs was significantly lower in the control leaves on day 8 of vase life ($p < 0.01$).

The patterns of MG activities showed an increasing trend ranging between 0.17 and 0.32 unit·mg$^{-1}$ protein during the vase period (Figure 4C). Leaves pulsed with BA and the control leaves had a higher activity of MG, at 0.31 and 0.32 unit·mg$^{-1}$ protein, respectively, on day 8 than that of leaves pulsed with TDZ, which slightly changed during the vase period ($p < 0.01$). Figure 4D shows the activity of PPH, which was expressed as approximately 0.0018–0.0046 unit·mg$^{-1}$ protein during the vase period. The declining trend of PPH activity was found in leaves pulsed with CKs. The activity of PPH in TDZ treatment was significantly lower, at 0.0018 unit·mg$^{-1}$ protein, than that of BA, which was expressed as 0.0028 unit·mg$^{-1}$ protein on day 8, while PPH activity of the control leaves was constant from the initial day to day 8, at 0.0044 unit·mg$^{-1}$ protein, then reduced afterwards ($p < 0.01$).

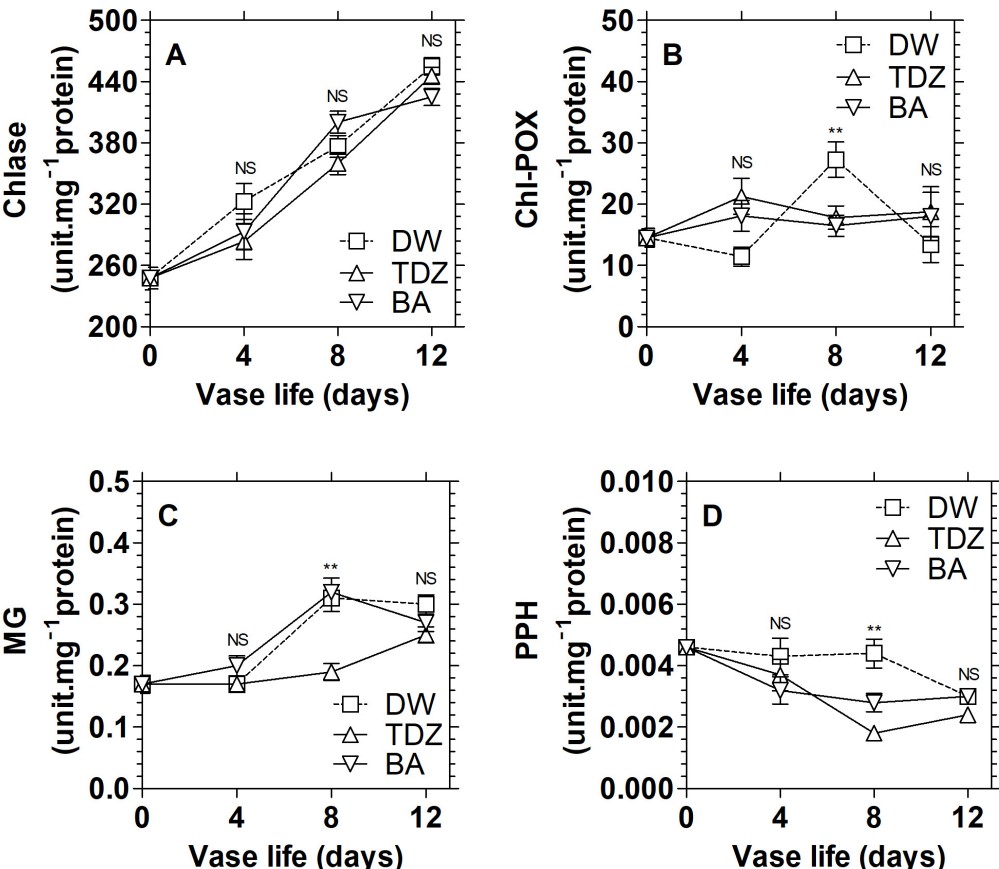

**Figure 4.** Effect of CKs on activities of chlorophyll catabolism enzymes; chlorophyllase; Chlase (**A**), chlorophyll peroxidase; Chl-POX (**B**), magnesium dechelatase; MG (**C**) and pheophytinese; and PPH (**D**) in *D. solida* leaves. Leaves were pulsed with distilled water (DW; control), 10 µM TDZ, 100 ppm BA for 24 h, then transferred to distilled water in a controlled environment room for the vase period. Asterisks above the curves at each time point represent significant differences between treatments according to Duncan's multiple range test (DMRT); $p < 0.01$ (**) and NS not significant.

*3.3. Effects of TDZ under MAP on the Display Quality of D. solida Leaves after Storage*

As a result of experiment 1, treatment with 10 µM TDZ was selected to evaluate display quality after a long storage period and during the vase period, compared with the control leaves. After pulsing, leaves were packed in BOPP bag, and then stored at 10 °C under light conditions for 12 h/day for one, two and three weeks. The leaves were transferred to distilled water and monitored for their post-storage life in the vase. During storage under MA, $O_2$ levels seemed constant, while $CO_2$ accumulated to high levels (0.58%) during the first week, then remained stable afterwards (data not shown). It was found that one-week storage gave better results in maintaining the Chl content in leaves than other storage durations (Figure 5A–C). The contents of total Chl and Chl *a* were 72.22 and 48.98 mg·100 g$^{-1}$ FW, respectively, prior to storage. After one-, two- and three-week storage, the leaves had a higher total Chl content of 83.15, 82.70 and 86.09 mg·100 g$^{-1}$ FW and a Chl *a* content of 56.28, 56.66 and 58.52 mg·100 g$^{-1}$ FW, respectively; then the contents of total Chl and Chl *a* continuously declined to 52.32, 42.75 and 43.11 mg·100 g$^{-1}$ FW and to 35.08, 29.01 and 29.04 mg·100 g$^{-1}$ FW, respectively, on day 12 of the vase period. Moreover, the Chl *b* content was 15.21 mg·100 g$^{-1}$ FW prior to storage. After one-, two- and three-week storage, the content of Chl *b* ranged between approximately 83.15, 82.70 and 86.09 mg·100 g$^{-1}$ FW, then declined to 52.32, 42.75 and 43.11 mg·100 g$^{-1}$ FW on day 12. TDZ significantly delayed the reduction of Chl content in one-week storage. It was found that pulsing solutions significantly affected the content of total Chl and Chl *a* during days

4–8 of the vase period ($p < 0.01$). In addition, a significant difference in the relationship between the pulsing solutions and storage duration was observed ($p < 0.01$).

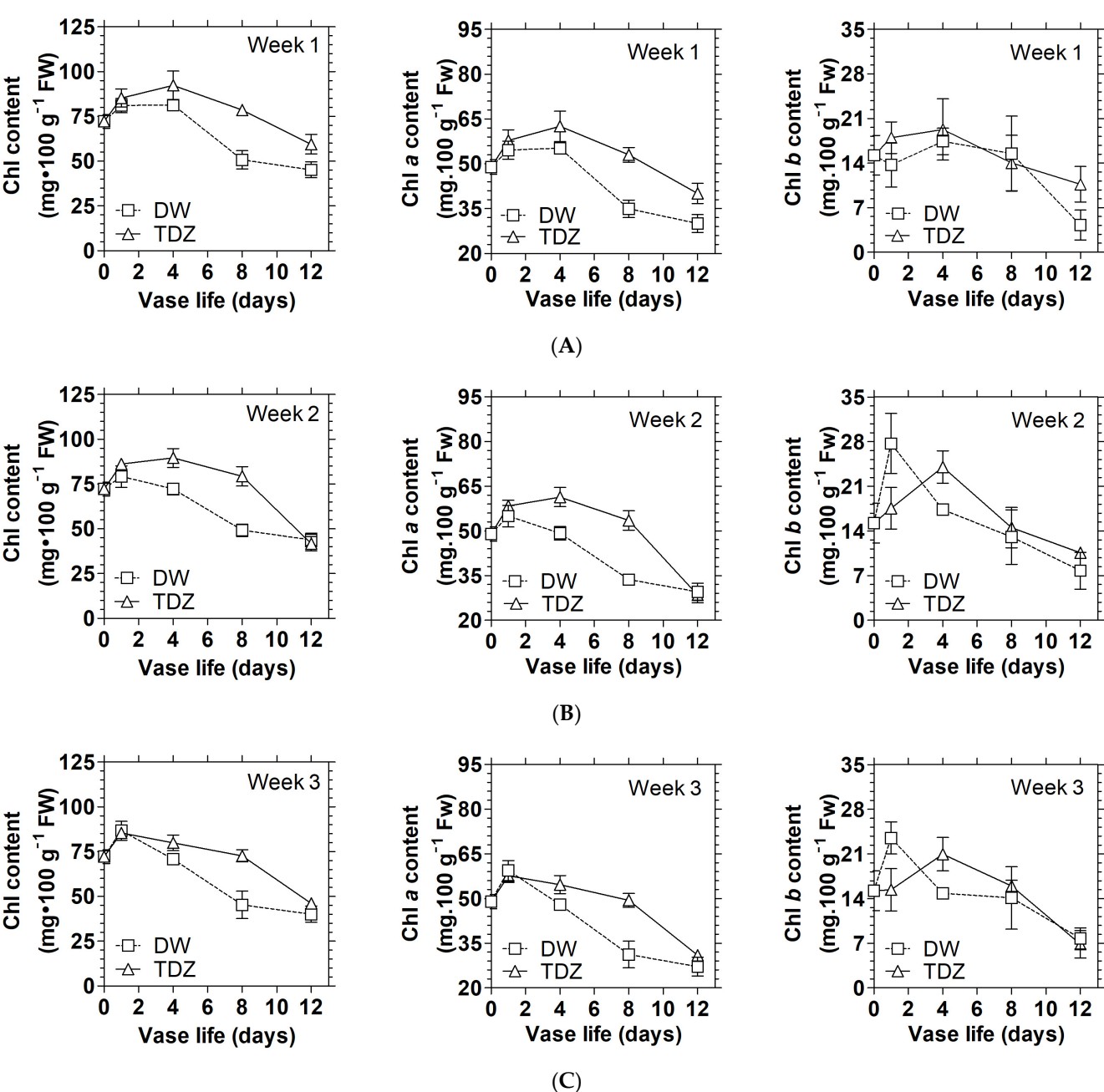

**Figure 5.** Effect of TDZ and MAP on the contents of total Chl (**A**), Chl *a* (**B**) and Chl *b* (**C**) in *D. solida* leaves after storage. Leaves were pulsed with 10 µM TDZ and distilled water (DW; control), then packed in BOPP bag, and stored at 10 °C under cool-white fluorescent lights for a 12-h photoperiod for 1, 2 and 3 weeks. After storage, ferns were transferred to distilled water in a controlled environment room for the vase period.

Leaf yellowing after one-, two- and three-week storage ranged between approximately 8, 10 and 16%, respectively, and then continuously rose throughout the vase period (Figure 6). On day 12, the yellowing of the leaves was 69, 71 and 85%, respectively. The result revealed that cut leaves stored for one week had a lower percentage of leaf yellowing than the two- and three-week storage; and that pulsing treatment with TDZ significantly

delayed the yellowing of leaves to 57% after one-week storage, while that of the control and TDZ treatment groups were 85% after three-week storage.

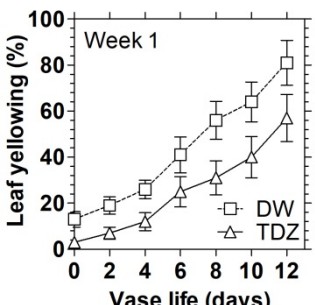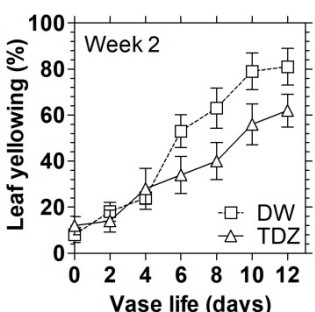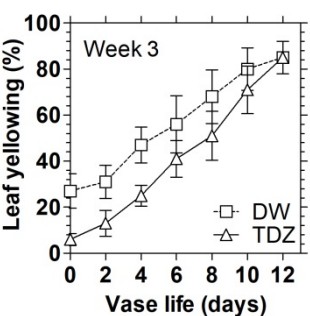

**Figure 6.** Changes in yellowing of *D. solida* leaves pulsed with 10 μM TDZ or distilled water (DW; control) for 24 h, packed in a BOPP bag, stored at 10 °C for 1 (**Left**), 2 (**Middle**) and 3 weeks (**Right**), and then transferred to distilled water in a controlled environment room.

Table 1 shows that pulsing treatment with TDZ combined with one-week storage significantly prolonged the vase life of leaves to 9.6 days, following by that of TDZ with two- and three-week storage, which had 7.6 and 6.8 days of vase period, respectively, while the control leaves stored for three weeks had the shortest vase life of 3.6 days. It was found that pulsing solutions and storage duration significantly affected the percentage of leaf yellowing and vase life over the vase period ($p < 0.01$). Moreover, significant difference in the relationship between pulsing solutions and storage duration was observed ($p < 0.01$).

**Table 1.** Effect of TDZ and MAP on vase life of *D. solida* leaves after storage. Leaves were pulsed with 10 μM TDZ and distilled water (DW; control), then packed in BOPP bag, and stored at 10 °C under cool-white fluorescent lights for 12 h/d for 1, 2 and 3 weeks. After storage, ferns were transferred to distilled water in a controlled environment room for the vase period.

| Treatments | | Vase Life (Days) [1] |
|---|---|---|
| **Solution** | **Storage Period (Weeks)** | |
| DW | | 5.2 [b] |
| TDZ | | 8.0 [a] |
| F-test | | ** |
| | 1 | 7.9 [a] |
| | 2 | 6.7 [ab] |
| | 3 | 5.2 [b] |
| F-test | | * |
| DW | 1 | 6.2 [bc] |
| | 2 | 5.8 [bc] |
| | 3 | 3.6 [c] |
| TDZ | 1 | 9.6 [a] |
| | 2 | 7.6 [ab] |
| | 3 | 6.8 [abc] |
| F-test | | ** |

[1] Means within the same column with different letters represent significant differences between treatments according to Duncan's multiple range test (DMRT). Asterisks represent significant differences in variance at $p < 0.01$ (**) and $p < 0.05$ (*).

### 3.4. Effects of TDZ under MAP on the Display Quality of D. solida Leaves after Storage

The relative fresh weight of leaves and water uptake had a similar trend of decline after one week of storage (Figure 7A,B). The leaves of *D. solida* showed a gradual decrease

in relative fresh weight throughout the vase period. Moreover, on the initial day after storage, the water uptake of leaves pulsed with TDZ was the highest at 16.96 mL, while that of control fronds had 12.20 mL, then rapidly decreased over the vase period. No significant difference was observed in the fresh weight and water uptake between the control leaves and those treated with TDZ.

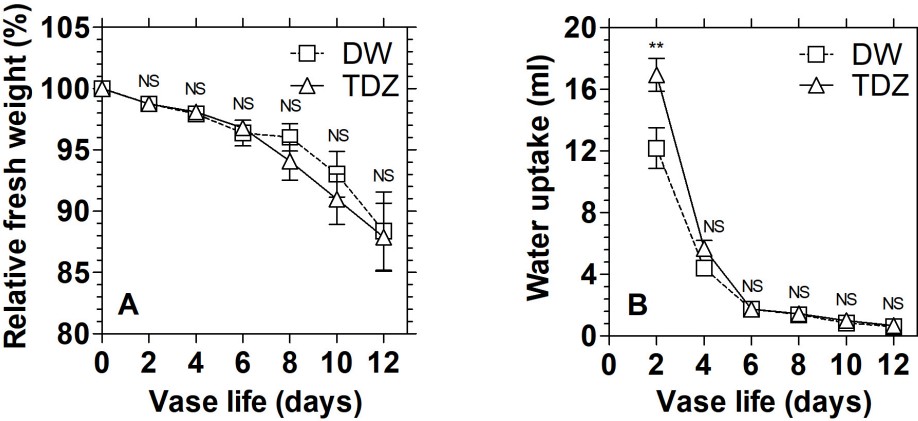

**Figure 7.** Effect of TDZ and MAP on fresh weight (**A**) and water uptake (**B**) of *D. solida* leaves after 1-week storage. Leaves were pulsed with 10 μM TDZ and distilled water (DW; control), then packed in BOPP bag, and stored at 10 °C under cool-white, fluorescent lights for 12 h/d for 1 week. After storage, ferns were transferred to distilled water in a controlled environment room for the vase period. Asterisks (*) indicate significant differences between the two treatments after storage (*t*-test; ** $p < 0.01$, NS not significant).

The contents of total Chl, Chl *a* and Chl *b* at zero time (before storage) ranged between approximately 70.38, 47.72 and 14.73 mg·100 g$^{-1}$ FW, respectively, and tended to decrease during the vase period (Figure 8A–C). The control leaves had lower contents of total Chl and Chl *a* to 38.17 and 25.87 mg·100 g$^{-1}$ FW on day 12 of vase life, while the leaves pulsed with TDZ maintained total Chl and Chl *a* contents at 49.48 and 33.53 mg·100 g$^{-1}$ FW, respectively. However, no significant difference was observed in Chl *b* content between the control leaves and those treated with TDZ.

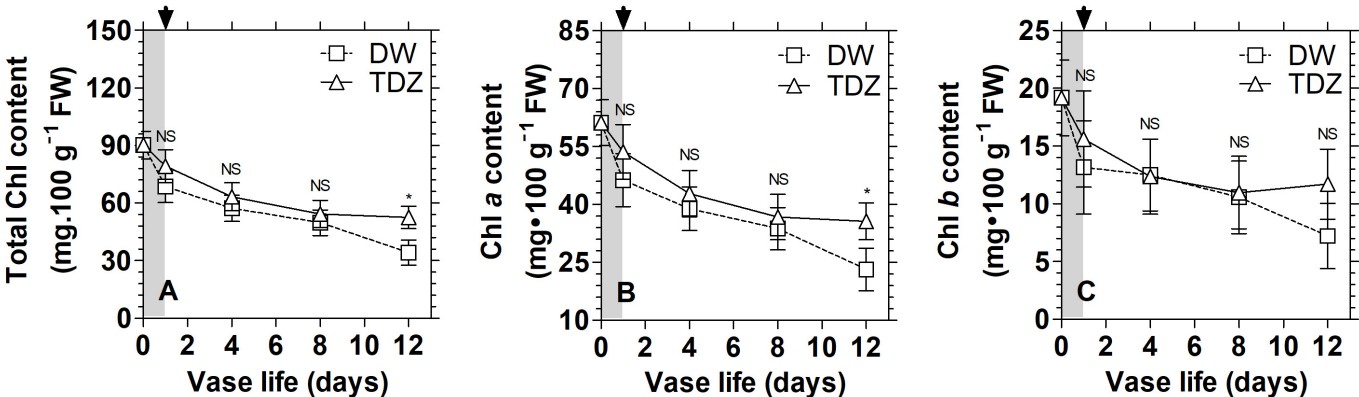

**Figure 8.** Effect of TDZ and MAP on the contents of total Chl (**A**), Chl *a* (**B**) and Chl *b* (**C**) in *D. solida* leaves after 1-week storage. Leaves were pulsed with 10 μM TDZ and distilled water (DW; control), then packed in BOPP bag, and stored at 10 °C under cool-white fluorescent lights for 12 h/d for 1 week. After storage, ferns were transferred to distilled water in a controlled environment room for the vase period. Asterisks (*) indicate significant differences between the two treatments after storage (*t*-test; * $p < 0.05$, *ns* not significant). The black arrow indicates the initial day in the vase.

The yellowing ranged between 4 and 9% in the control leaves and those treated with TDZ on the initial day after storage, and increased continuously to 73 and 63% leaf

yellowing, respectively, on day 12 of the vase period (Figure 9A). Treatment with TDZ pulsing prior to storage significantly delayed leaf yellowing during days 6–8, then extended the vase life of leaves to 9.1 days, longer than the control leaves, which had 7.1 days of vase life (Figure 9B).

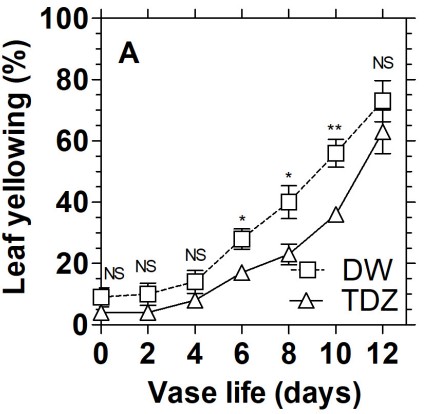 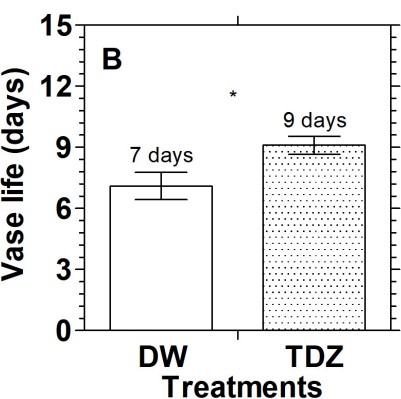

**Figure 9.** Effect of TDZ and MAP on yellowing of *D. solida* leaves after 1-week storage (**A**). Leaves were pulsed with 10 μM TDZ and distilled water (DW; control), then packed in BOPP bag, and stored at 10 °C under cool-white fluorescent lights for 12 h/d for 1 week. After storage, ferns were transferred to distilled water in a controlled environment room for the vase period (**B**). Asterisks (*) indicate significant differences between the two treatments after storage (*t*-test; * $p < 0.05$, ** $p < 0.01$, NS not significant).

### 3.5. Effects of TDZ under MAP on the Activities of Chlorophyll Catabolism Enzymes in D. solida Leaves after Storage

At zero time, the activity of Chlase was 274.83 unit·mg$^{-1}$ protein, then decreased to 181.03 unit·mg$^{-1}$ protein in the control leaves and 234.65 unit·mg$^{-1}$ protein in the TDZ pulsing treatment group after storage (Figure 10A). The Chlase activity in the control leaves increased to 304.09 unit·mg$^{-1}$ protein on day 12, which was found to be higher than that of TDZ. The leaves pulsed with TDZ showed a slight change in the activity of Chlase during the first 8 days, then sharply increased to 279.25 unit·mg$^{-1}$ protein on day 12 of vase life. The TDZ treatment significantly delayed the increase in Chlase activity during the vase period ($p < 0.01$).

The activity of Chl-POX was 7.38 unit·mg$^{-1}$ protein at zero time, and higher at 11.93 unit·mg$^{-1}$ protein in the control leaves and at 10.83 unit·mg$^{-1}$ protein in the TDZ pulsing treatment group after storage (Figure 10B). The trend of Chl-POX activity still increased throughout the vase period, but the leaves pulsed with TDZ had a lower activity of Chl-POX than the control. The results revealed that the activity of Chl-POX in the leaves pulsed with TDZ was 16.58 unit·mg$^{-1}$ protein, significantly lower than the control, which had 27.81 unit·mg$^{-1}$ protein on day 8 of vase life. However, no significant difference was observed in the activity of Chl-POX over the vase period.

MD activity in *D. solida* leaves was 0.34 unit·mg$^{-1}$ protein at time zero, then decreased to 0.25 unit·mg$^{-1}$ protein in the control leaves and to 0.23 unit·mg$^{-1}$ protein in the TDZ pulsing treatment group (Figure 10C). The activity of MD gradually increased to 0.30 and 0.35 unit·mg$^{-1}$ protein in the control and TDZ treatment group, respectively. The leaves pulsed with TDZ had significantly lower activity of MG than those in the control group ($p < 0.05$).

Figure 10D shows that PPH activity in *D. solida* leaves was approximately 0.0038 unit·mg$^{-1}$ protein at time zero. After 1-week storage, the activity of PPH in the control leaves was higher to 0.0047 unit·mg$^{-1}$ protein while that of TDZ treatment was 0.0035 unit·mg$^{-1}$ protein, than slowly declined to 0.0039 and 0.0024 unit·mg$^{-1}$ protein, respectively, on day 12 of the vase period. It was obvious that the activity of PPH in the control leaves was

significantly higher than that of the TDZ pulsing treatment group ($p < 0.01$) throughout the vase period.

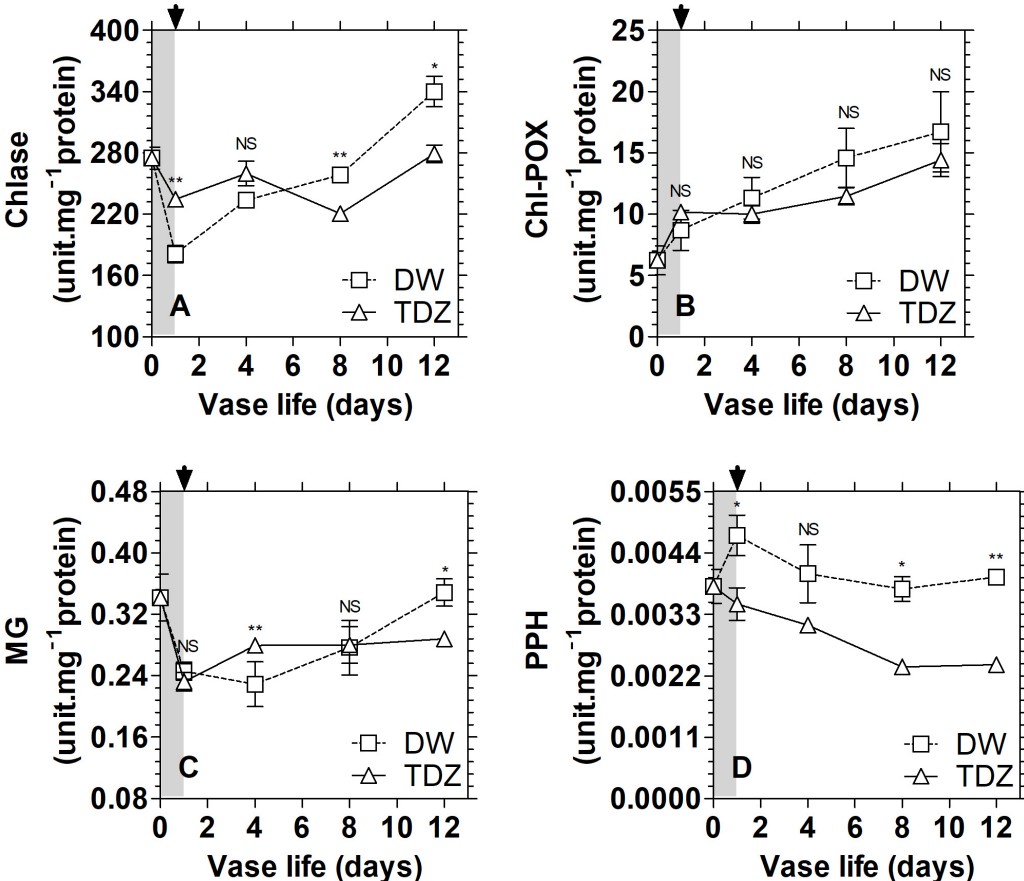

**Figure 10.** Effect of TDZ and MAP on activities of Chl catabolism enzymes; chlorophyllase; Chlase (**A**), chlorophyll peroxidase; Chl-POX (**B**), magnesium dechelatase; MG (**C**) and pheophytinese; PPH (**D**) in *D. solida* leaves after 1-week storage. Leaves were pulsed with 10 µM TDZ and distilled water (DW; control), then packed in BOPP bag, and stored at 10 °C under cool-white fluorescent lights for 12 h/d for 1 week. After storage, ferns were transferred to distilled water in a controlled environment room for the vase period. Asterisks (*) indicate significant differences between the two treatments after storage (*t*-test; * $p < 0.05$, ** $p < 0.01$, *ns* not significant). The black arrow indicates the initial day in the vase.

## 4. Discussion

The yellowing of leaves is the most obvious visible senescence symptom resulting from the degradation of Chl and other components of chloroplasts that occurs in most leafy plants, including *D. solida*. Previous studies have reported that leaf senescence is also accompanied by the breakdown of Chl *a* and *b* [55], which involves non-enzymatic mechanisms [56,57] and multistep enzymatic mechanisms [52]. The former is associated with an increase in levels of reactive oxygen species (ROS) in leaves during senescence [57,58], and the latter is directly caused by the activities of Chl-degrading enzymes such as chlorophyllase, Mg-dechelatase, pheophorbide *a* oxygenase and red chlorophyll catabolite reductase [52]. The study by Teerarak and Laosinwattana [24] revealed that a 6-h pulse treatment with 25 mg L$^{-1}$ ginger essential oil in combination with holding treatment in 5 mg L$^{-1}$ ginger essential oil could prolong vase life and enhance postharvest quality of *D. solida* leaves by simultaneously retarding the degradation of Chl *a* and *b*, maintaining DPPH scavenging activity and reducing the accumulation of MDA content. Moreover, the senescence of leaves is regulated by PGRs, which significantly delay senescence and maintain Chl content. Alterations in the level of CKs are thought to be involved to the

initiation of the senescence process of leaves. Exogenous applications of BA at concentrations from 2 to 200 μM for 24 h or 5–6 days almost completely prevented senescence [59]. Previous studies showed that the exogenous application of CKs or the transgenic expression of CK biosynthesis genes prevented the degradation of Chl, photosynthetic proteins and RNA, which resulted in delayed senescence [60–62], while CK biosynthesis mutants have a shorter leaf lifespan [63]. Thidiazuron (TDZ), a market-available cytokinin-like compound, has been reported to have high CK activities at concentrations lower than other adenine-type CKs [36,64]. Our results showed that pulsing treatments with CKs gave better results than the control in preventing leaf yellowing and extended the vase life of *D. solida* leaves, but TDZ significantly maintained the content of total Chl, which is probably caused mainly by its greater inhibiting of MG activity in the leaves when compared with BA. This finding is supported by the previous study by Tatmala et al. [5], who reported that holding 10 μM TDZ significantly extended the vase life of cut *D. solida* leaves longer than the control. TDZ increased the level of Chl in five cultivars of *Pelargonium zonale* hybrid cutting, while the level of Chl declined in untreated controls [65]. Genkov and Ivanova [38] reported that TDZ demonstrated higher activity in *Dianthus* micropropagation than BA at concentrations 100 times lower than those optimal for BA. The phenylurea CKs showed typical CK effects, such as increased shoot multiplication and higher Chl content. In cut *Alstroemeria*, leaf yellowing can be delayed either by a continuous treatment with 1 μM TDZ or by a pulse treatment with 10 μM TDZ for 24 h [40]. The application of TDZ prolonged vase life by inhibiting leaf yellowing, but simultaneously enhanced ethylene production in cut *Matthiola incana* flowers [66]. During leaf senescence, Chl molecules are degraded, whereas Chl *b* is converted to Chl *a* by Chl *b* reductase (CBR) [67,68]. Chlase is involved in the first step of the Chl catabolic pathway, which catalyzes the conversion of Chl *a* to chlide *a* and phytol [53]. The activity of Chlase tends to decrease with senescence progress [69,70]. In this study, the activity of Chlase was found to be high and continuously increased in both control and cytokinin-treated leaves until the end of vase life. This indicated that CKs had no effect on Chlase activity in the yellowing process of fern leaves. Peroxidase is also involved in the Chl degradation pathway. In the presence of $H_2O_2$ and phenolic compounds with a hydroxyl group at the *p*-position, such as *p*-coumaric acid and apigenin, Chl *a* is also converted to $13^2$-hydroxychlorophyll a, a fluorescent chlorophyll catabolite, by Chl-POX [20,71]. It was found that Chl-POX activity in CK treated leaves was rather stable during the vase period, while that of the control leaves significantly increased on day 8, which was almost the end of vase life. Chlide *a* is converted to pheophytin *a* (Phein a) by MG, which was found to be the second step in the pathway of Chl degradation which removes magnesium from Chlide *a* [72]. Phein *a* is then hydrolyzed by pheophytinase (PPH) to produce pheophorbide *a* (Pheide *a*) and phytol [22,73]. TDZ treatment significantly delayed the increase in MG activity and reduced the activity of PPH as compared with the control, while BA induced only the reduction of PPH activity. This suggests that MG activity may be a rate-limiting step in Chl degradation, which is one feature that sets TDZ apart from BA. TDZ could significantly delay leaf yellowing in *D. solida* by retarding Chl degradation via the suppression of Chl-POX, MG and PPH activities on day 8 of vase life. Our study confirmed that the application of TDZ was more effective than BA in maintaining the quality of leafy plants. The physiological efficiency of TDZ on plants might be due to its cytokinin-like activity, which is higher than BA. Additionally, TDZ is not metabolized by plants, resulting in longer activity than other CK compounds [38]. TDZ might promote the conversion of cytokinin ribonucleotides to more biologically active ribonucleosides [74].

This study also varied the duration of the storage period after pulsing treatment with TDZ, compared with the control. The storage conditions were at 10 °C under light conditions for 12 h/day to prevent leaf yellowing. The recommendation for the storage temperature of ferns should be 4–4.5 °C, which can maintain the vase life of ferns species such as leatherleaf fern for up to one month [75], while for *Nephrolepis exaltata* leaves the vase life was still acceptable at 16 days [76]. In this study, we found that 10 °C storage was suitable for cut tropical ferns like *D. solida*. The storage duration in MAP affected

the vase life of the pulsed ferns. Previous studies reported that in dark conditions, TDZ was less effective in the suppression of Chl degradation. This effect might be due to its inactivation or by the lack of light in the Chl turnover [64]. Moreover, the biosynthesis of Chl in the last step mediated by NADH protochlorophyllide reductase, which converts the protochlorophillide to chlorophyllide, requires light. Light regulates Chl biosynthesis at transcriptional level, while the CKs act at the level of post-transcription [77–79]. Wheat plants grown in dark conditions showed a 10-fold lower amount of CKs than those grown in light conditions [66]. TDZ pulsing treatment significantly maintained the decreased contents of total Chl, Chl *a* and *b* as compared with the control after storage, and one-week storage gave the best results in delaying leaf yellowing, extending the vase life to approximately 10 days. The vase life of cut *D. solida* leaves was shorter with a storage period. A three-week storage period of control leaves had the shortest vase life of 3.6 days, but application of TDZ obviously doubled the vase life of the leaves. Therefore, the most suitable storage conditions for *D. solida* leaves were pulsing with TDZ for 2 h prior to storage for one week. However, TDZ pulsing treatment did not affect the relative fresh weight and water uptake after storage, although it did stimulate the highest uptake of solution on the initial day after storage. Moreover, treatment with TDZ led to a higher content of total Chl, Chl *a* and *b* than in the control leaves, resulting in delaying the yellowing of leaves and longer vase life of 9.0 days, while the control had shorter vase life of 7 days. The pattern of activities of Chl catabolic enzyme had a similar trend to that of Chl degrading activity before storage. In addition, pulsing with TDZ prior to storage could significantly suppress or delay the increase in the Chl catabolic enzymes Chlase, Chl-POX, MG and PPH during the vase period, which maintains the display quality of cut leaves of *D. solida*.

## 5. Conclusions

The yellowing of *D. solida* leaves can be best avoided by using pulse treatments with 10 μM TDZ during the vase period or storage period. We suggest that one-week storage under MA with light conditions for cut *D. solida* leaves could significantly maintain the postharvest quality after storage. Moreover, TDZ pulsing treatment effectively delayed Chl-degrading activities such as Chlase, Chl-POX, MG and PPH, indicating that the suppression of those enzyme activities by TDZ treatment could be involved in retarding Chl degrading enzymes during storage and vase periods. This study provides practical benefits, and could be applied by the producers of ferns to improve the quality of *D. solida* leaves after harvest.

**Author Contributions:** Conceptualization, M.B., V.S. and C.W.-A.; Methodology, P.N.; formal analysis, P.N.; investigation, P.N.; writing—original draft preparation, M.B., V.S. and C.W.-A.; writing—review and editing, M.B. and V.S.; supervision, M.B. and C.W.-A.; visualization, M.B., V.S. and C.W.-A. All authors have read and agreed to the published version of the manuscript.

**Funding:** This research did not receive any funding.

**Institutional Review Board Statement:** Not applicable.

**Informed Consent Statement:** Not applicable.

**Data Availability Statement:** Data sharing is not applicable to this article.

**Acknowledgments:** The authors would like to thank The Postharvest Technology Innovation Center, Ministry of Higher Education, Science, Research and Innovation, Bangkok and The United Graduated School of Agricultural Science (UGSAS), Gifu University, Japan for supporting some pieces of scientific equipment in this study.

**Conflicts of Interest:** The authors declare no conflict of interest.

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
