# Peer review of "Application of Cytokinin under Modified Atmosphere (MA) Delays Yellowing and Prolongs the Vase Life of Davallia solida (G. Forst.) Sw. Leaves"

_agriculture, doi:10.3390/agriculture13020463_

Round 1
Reviewer 1 Report
The authors performed a valuable investigation of the effect of cytokinin on vase life of Rabbit’s foot fern (Davallia solida) leaves. The strengths of the article include the object of study, which is a little-known fern, and in-depth biochemical analyses of chlorophyll degradation. The weakness of the article are one-time results.
Below are my comments:
Abstract
Davallia is the name of the genus. In this study evaluated a specific species (so it appears from the title) so you should write Davallia solida the first time, and then throughout the paper write D. solida.
Why is chlorophyll written with a capital letter?
What does the abbreviation BOPP stand for?
The abstract should be completed by the main conclusions.
Keywords
The words of the title should not be repeated.
Introduction
The introduction needs a thorough revision to emphasize the innovative research.
The current state of the research field should be supplemented with data on the global business of cut foliage industry, the role of ferns in the offer of cut foliage cut, the botanical description of the species studied, systematic affiliation, the scale of production of D. solida in Thailand and the world, the storage of cut foliage/ferns based on specific studies, the effect of hormones on the longevity of cut foliage/ferns.
Are the described biochemical processes involved in chlorophyll degradation based on studies in seed plants or ferns? Are there such differences?
I consider it a mistake to refer to the results of studies relating to changes in the flowers of anthuriums, heliconias, ginger, mokara orchids, phlox or lupines
Lack of any consistency in the writing of plant names - once they are common names, once they are species names, and once they are the name of the genus itself, e.g. Ornithogalum - but it is not clear what species or variety it is.
According to MDPI guidelines, the introduction should including specific hypotheses being tested. Unfortunately, the authors do not present a research hypothesis.
Materials and Methods
Plant material. What it means "the commercial stage"? There must be photos. No data on growing conditions, how many days it was from cutting the leaves? Whether the leaves were conditioned at the grower? Whether the leaves had spores?
Experiments. The description of the experience is vague and chaotic. The authors in the paper use the terms experiment 1 and 3 and treatment 2. If there are 3 experiences then these 3 experiences must be described separately. Whether the leaves were inserted individually into the tubs? What was the volume (ml)? Which exact parts of the leaf blade were taken for analysis, how many samples were taken from one leaf, did these samples have spores below?
Line 134 and 135.The authors also studied spinach leaves???
Results
Description of results should be more concise and clearer.
On charts 1, 2, 3, there are no letters next to the averages (homogeneous groups) and such information is next to the description of the chart. DW, TDZ and BA explanations are missing from the charts: 1A, 1B, 3B, 3C, 3D, 4A, 4B, 4C, 5 week 2, 5 week 3, 6A?
Discussion
Missing from the discussion is a broader view of fern leaf storage of other species. Future research directions may also be mentioned.
Why the authors omit such an important item?: Teerarak, M., & Laosinwattana, C. (2019). Essential oil from ginger as a novel agent in delaying senescence of cut fronds of the fern (Davallia solida (G. Forst.) Sw.). Postharvest Biology and Technology, 156, 110927.
So valuable research is at work: Allen, T. B. (1997). A fine structural analysis of chloroplast developmental stages in the fern Davallia fejeensis with comparisons to chloroplasts of other major plant groups. Teachers College, Columbia University.
Author Response
Thank you very much for your value comments, really appreciate!

Reviewer 2 Report
I have various concerns before the manuscript can be reconsidered for publication in Agriculture . In the current article, the authors studied Manuscript entitled "Application of cytokinin under modified atmosphere (MA) delays leaf yellowing and prolongs the display life of rabbit’s foot fern (Davallia solida (G. Forst) Sw.)
Abstract: In abstract (P<0.05) should be written as P=0.05
Line 12-16 is too long to read. Kindly rephrase the sentences.
The manuscript needs English language (grammar) improvement through out the manuscript.
What are the recommendations? Mention briefly in the abstract section.
Keywords: Keywords written should be different from the title section.
Introduction: Importance of cytokinin as preservative solution should be mentioned.
Line 68 why the example of red and pink ginger is mentioned. I suggest to delete this example and add some related examples.
Material and methods:
Line 86 mention the commercial stage of fern
Line 95 check RH as it is different from the RH mentioned in the abstract section line 15
When the experiment was conducted? Mention it.
In line 97 correct 21+2 ℃ as 21±2 ℃ throughout the manuscript.
Line 108 mention how to calculate water uptake
Line 120 write 4℃ as 4 ℃. Follow same pattern throughout the manuscript.
Line 126 Change preparation of substrates to preparation of content.
Line 139 mention hour as h as it is mentioned earlier. Correct it throughout the manuscript.
Line 144 correct five hundred as 500.
Line 159-160 expand MG, Chl-POX and PPH
Line 192 write Hcl as HCl
Line 200 write experiment instead of treatment.
Results
Line 209 check this range 42.73 to 65.57
Line 259 check the significance level of text and graph some where it is mentioned as (P<00.5 or P<00.1). Check this through out the manuscript.
Line 261-264 rephrase the sentence.
Line 266-268 rephrase the sentences
Line 337 16.96 ml was highest in which treatment check and rewrite.
Discussion
Needs major improvement.
Support with relevant crops
Conclusion
Needs improvement with some future prospects.
Figures
Fig. 2 Write leaf as Leaf; vase as Vase
Fig. 8 Write treatments at the bottom of DW and TDZ in Fig. B
Author Response

(The authors gave the same response as above.)

Round 2
Reviewer 1 Report
The authors significantly improved the article using the comments. A photo of the leaves of the studied fern is missing, for example, in the introduction. The species under evaluation is little known and the addition of a photo of it will increase interest in the research results presented
Author Response
Thank you very much for your hard work, I'm so thankful!
Reviewer 2 Report
The authors responded appropriately to the referees' comments, however, some minor changes need to be addressed. After the addition of these minor changes the manuscript can be accepted.
Author Response

(The authors gave the same response as above.)
